# Relationship between Type and Intensity of Sports Activities and the Prevalence of Overweight in Serbian School Children

**Sead Malićević** [1] , **Sanja Mazić** [2], **Stefania Cataldi** [3,*] , **Francesco Fischetti** [3,†] and **Gianpiero Greco** [3,†]

1 Department of Sports Medicine, College of Sports and Health, 11010 Belgrade, Serbia; sead.malicevic@vss.edu.rs
2 Institute of Medical Physiology, School of Medicine, University of Belgrade, 11000 Belgrade, Serbia; sanja.mazic@med.bg.ac.rs
3 Department of Basic Medical Sciences, Neuroscience and Sense Organs, University of Study of Bari, 70124 Bari, Italy; francesco.fischetti@uniba.it (F.F.); gianpiero.greco@uniba.it (G.G.)
* Correspondence: stefania.cataldi@uniba.it
† These authors share last authorship.

**Abstract:** This study aimed to explore the relationship between different forms and intensities of sports activities and the prevalence of overweight in primary school children. Using International Obesity Task Force BMI criteria, we have identified body weight status in a group of 2893 children aged 9–15 years participating in 27 different sports and within a group of 4987 non-active children of the same age. We have compared the prevalence of overweight and obesity between these groups, as well as between genders and different forms and intensities of sports, within the group of active children. We have found lower prevalence of overweight ($X^2$ = 41.689, DF = 1, $p < 0.001$), as well as prevalence of obesity ($X^2$ = 175.184, DF = 1, $p < 0.001$) in physically active children compared with their non-active counterparts, as expected. Overweight (including obesity) had the highest prevalence in mixed sports of the Classification of Sports of the European Association of Preventive Cardiology, in boys ($p = 0.003$), as well as in girls ($p = 0.043$). A lower prevalence of overweight was noted in boys ($p = 0.001$), as well as in girls ($p = 0.025$) with more than 7 years of training. Regarding the number of hours of training per week, a lower prevalence of overweight was noted only in girls with 4 and more hours of training per week ($p = 0.025$). Concerning intensity, we have found a significant drop in the prevalence of overweight in children with sports activities whose intensity is more than 60 MET-hours per week. We conclude that a significant relationship between sports and the prevalence of overweight is found in children with more than 60 MET-hours of sports activities per week, as well as among children involved in mixed sports, and after more than 7 years spent in regular sports training.

**Keywords:** children overweight and obesity; youth sports; sustainable prevention of overweight and obesity; Youth Compendium of Physical Activities

## 1. Introduction

In this paper, we use the term overweight for subjects with overweight and obesity, and the term obesity for subjects with obesity alone.

The Non-communicable Disease Risk Factor Collaboration (NCD RisC) meta-analysis estimated that there are around 213 million overweight and 124 million children and adolescents (age 5–19) with obesity in the World [1]. The prevalence of overweight among children and adolescents aged 5–19 has risen dramatically from just 4% in 1975 to over 18% (18% of girls and 19% of boys) in 2016 [1]. While only 0.8% of children and adolescents aged 5–19 (5 million girls and 6 million boys) was with obesity in 1975, more than 124 million children and adolescents (5.6% of girls and 7.8% of boys) were with obesity in 2016 [1]. Similarly, according to the latest official data from The Institute of Public Health of Serbia, 29.5% of overweight children (12.9% of which with obesity), were identified in 2019 in

Serbia [2]. One recent Serbian national representative study revealed that 24.2% of school children are overweight, 5% of whom are with obesity [3]. These data are particularly worrying since being overweight in childhood and adolescence leads to overweight and/or overweight-related health problems in later life [4–6]. Besides that, being overweight in childhood may affect motor competence for physical activities, as well as for professional activities in later life. Therefore, early involvement in sports activities may be important to counteract weight-related negative effects, through the improvement of movement skills and cognitive functions [7,8].

While numerous studies have investigated the relationship between physical activity and overweight in children, most of them were focused only on general or school-based physical activity as an anti-obesogenic factor in children [9–12]. Although important, the prevalence reported in these studies is a result of both contrasting groups of children: those physically active, and those whose activity has been limited only to physical education school classes. Surprisingly, not so many studies [13–24] have explored the potential relationship between physical activities such as sports and being overweight. Of particular importance for this study is that to date no data on the potential difference in the prevalence of overweight among children taking part in various sports activities in Serbia has been reported.

Information about the prevalence of overweight in children engaged in sports would help to supply a more objective inside on the relationship between participation in sports activities during childhood and body status and may provide support to the national strategy on sustainable prevention of non-communicable diseases.

This study aimed to investigate the body weight status of physically active and non-active school children the relationship between the type and intensity of sports activities, and the prevalence of overweight in Serbian school children actively involved in sports activities.

We hypothesized that a higher prevalence of overweight will be found:

-       in the group of non-active children,
-       in children engaged in power sports,
-       among children with less than 4 h of sports activities per week,
-       within children that are involved in sports activities for less than two years.

## 2. Materials and Methods

### 2.1. Participants and Procedures

For the group of active children, we have evaluated more than 3500 girls and boys between May and October 2018, in the Institute of Medical Physiology of the School of Medicine, during regular systematic medical check-ups (by the Law on Sports, in Serbia every person involved in organized sports activities must undergo medical examinations every six months).

Among all of those children, we were looking for specific candidates fulfilling the following criteria:

-       engagement in sports activities for at least 2 years,
-       minimum of 3 h of activity per week (in addition to 2 classes of 45 min per week of physical education school classes).

Originally, we wanted to evaluate primary school children, aged 7 to 15, but we have found very few children aged 7 and 8 years to qualify the abovementioned criteria, so we set the age of the participants to 9–15 years.

So, a total of 2893 physically active primary school children (1382 girls and 1511 boys) qualified to take part in this study group. They were engaged in 27 different sports or sports disciplines.

For the control group, we have assessed the body status of more than 7000 children of the same age in 17 primary schools in Belgrade during classes of physical education in May and June of 2018. The only criterium for inclusion in this group was the statement that

they were not involved in sports activities (i.e., physical education school classes were their only organized physical activity). So, after exclusion of children declaring participation in sports activities, 4987 non-active primary school children of the same age (2472 girls and 2515 boys) qualified to take part in the control group.

We have sorted all the active participants into one of four groups of sports according to the Classification of Sports of the European Association of Preventive Cardiology (EAPC): power sports, endurance sports, skill sports, and mixed sports [25] (Table 1).

**Table 1.** Distribution of Participants by Gender and Sports/Sports Disciplines in the Classification of Sports of the European Association of Preventive Cardiology (EAPC).

| Sports Group | Sports/Sports Discipline | Number of Subjects | | Total |
|---|---|---|---|---|
| | | Boys | Girls | |
| Skill Sports | Karate | 154 | 90 | 244 |
| | Dance | 5 | 28 | 33 |
| | Real Aikido | 14 | 9 | 23 |
| | Rhythmic Gymnastics | 0 | 4 | 4 |
| | Table Tennis | 19 | 5 | 24 |
| | Taekwondo | 15 | 12 | 27 |
| | Total Skill Sports | 207 | 148 | 355 |
| Power Sports | Athletics (pitching disciplines) | 3 | 6 | 9 |
| | Sports Gymnastics | 3 | 8 | 11 |
| | Kickboxing | 11 | 2 | 13 |
| | Wrestling | 13 | 0 | 13 |
| | Judo | 23 | 6 | 29 |
| | Total Power Sports | 53 | 22 | 75 |
| Endurance Sports | Athletics (running on long tracks) | 3 | 6 | 9 |
| | Athletics (running on middle tracks) | 1 | 7 | 8 |
| | Cycling | 5 | 0 | 5 |
| | Rowing | 6 | 0 | 6 |
| | Swimming | 150 | 142 | 292 |
| | Synchronized Swimming | 0 | 21 | 21 |
| | Total Endurance Sports | 165 | 188 | 353 |
| Mixed Sports | Water Polo | 47 | 7 | 54 |
| | Basketball | 483 | 39 | 522 |
| | Fencing | 8 | 4 | 12 |
| | Volleyball | 147 | 918 | 1065 |
| | Rugby | 11 | 0 | 11 |
| | Handball | 35 | 31 | 66 |
| | Tennis | 38 | 21 | 59 |
| | Football (Soccer) | 304 | 4 | 308 |
| | Ice Hockey | 13 | 0 | 13 |
| | Figure Skating | 0 | 12 | 12 |
| | Total Mixed Sports | 1086 | 1024 | 2110 |
| | Overall | 1511 | 1382 | 2893 |

Finally, we have expressed all the activities in energy expenditure (in MET–Metabolic Equivalent of Task) using the Youth Compendium of Physical Activities (YMET) [26,27] as we wanted to explore the relationship between the intensity of sports activities and the prevalence of overweight in our subjects. We have multiplied appropriate values in MET by the number of hours of activities per week for each of our subjects to express the intensity of their physical activity in MET-hours/week (MET-h/w).

### 2.2. Legal Issues

The parents or legal guardians gave written consent in the name of all participants. The study protocol adhered to the tenets of the Declaration of Helsinki and the Ethical Committee of the School of Medicine approved it (2650/IV-5, dated 10 April 2018).

### 2.3. Measurements

Experienced measurers performed all measurements following standardized procedures (with subjects being barefoot and in underwear).

We measured body weight using InBody 370 electronic scale (InBody Co., Ltd., Seoul, Korea), to the nearest 0.1 kg.

For body height, we used telescopic stadiometer Seca SE213 (Seca GmbH & Co. KG, Hamburg, Germany) to the nearest 0.1 cm.

Body mass index (BMI) was calculated as body weight divided by the square of body height and expressed in $kg/m^2$.

We have identified the body weight status of children based on their BMI values using the International Obesity Task Force (IOTF) criteria [28].

### 2.4. Statistical Analysis

Data are presented as absolute (n) and relative numbers (%). Differences in the prevalence of overweight and obesity in diverse groups, according to gender, EAPC sports categories, hours of training per week, number of years of training, and intensity of training were evaluated using the chi-square test. Statistical analyses were performed using the software package SPSS (IBM SPSS version 20.0, Chicago, IL, USA). The significance level was set at $p < 0.05$.

## 3. Results

A total of 2893 active and 4987 non-active children were included in this study. Their gender and body weight status distributions are given in Table 2.

**Table 2.** Distribution of Subjects by Gender and Body weight status.

|  |  | Active n (%) | Non-Active n (%) |
|---|---|---|---|
| Gender | Girls | 1382 (47.8) | 2472 (49.6) |
|  | Boys | 1511 (52.2) | 2515 (50.4) |
| Body weight status | Underweight | 175 (6.0) | 366 (7.3) |
|  | Normal Weight | 2069 (71.5) | 2761 (55.4) |
|  | Overweight * | 569 (19.7) | 1301 (26.1) |
|  | Obesity | 80 (2.8) | 559 (11.2) |

* In this table overweight does not include obesity.

We have found a lower prevalence of overweight ($X^2 = 41.689$, DF = 1, $p < 0.001$), as well as prevalence of obesity ($X^2 = 175.184$, DF = 1, $p < 0.001$) in physically active children compared with their non-active counterparts.

In the group of active children, a comparison of the prevalence of overweight (including obesity) between boys and girls, chi-square revealed a higher prevalence in boys than in girls (24.9% vs. 19.8%; $p < 0.01$). However, when the prevalence of obesity alone was

compared, gender differences between boys and girls were not significant: 3.0% vs. 2.5%, $p = 0.339$.

Next, as we wanted to investigate the relationship between the type of sports and body weight status, we have divided children by the type of physical activity (EAPC sports classification), the number of years of training, and the number of hours of training per week (Table 3).

**Table 3.** Overweight (including Obesity) and Obesity by Types of Activity, and Quantity of Training, descriptive.

| | | Number of Subjects | | Overweight n (%) | |
|---|---|---|---|---|---|
| | | **Boys** | **Girls** | **Boys** | **Girls** |
| EAPC Classification of Sports | Skill | 207 | 148 | 31 (15.0) | 25 (16.9) |
| | Power | 53 | 22 | 14 (26.4) | 5 (11.6) |
| | Endurance | 165 | 188 | 38 (23.0) | 21 (13.5) |
| | Mixed | 1086 | 1024 | 293 (27.0) | 222 (21.4) |
| Years of Training | 2–4 | 1067 | 1083 | 290 (27.2) | 225 (20.8) |
| | 5–7 | 367 | 275 | 78 (21.3) | 47 (17.1) |
| | more than 7 | 76 | 25 | 8 (10.5) | 1 (4.0) |
| Hours of Training per Week | 3 | 780 | 923 | 202 (25.9) | 201 (21.8) |
| | 4–5 | 592 | 395 | 144 (24.3) | 61 (15.4) |
| | 6 and more | 137 | 65 | 30 (21.9) | 11 (16.9) |

Overweight had the highest prevalence in mixed sports of EAPC (boys: $p = 0.003$, girls: $p = 0.043$), and lower prevalence in children with more than 7 years of training (boys: $p = 0.001$, girls: $p = 0.025$). Regarding the number of hours of training per week, a lower prevalence of overweight was noted only in girls having 4–5 h of sports activities per week ($p = 0.025$).

When the relationship between the intensity of sports activities expressed in MET-hours per week (MET-h/w) and body weight status was analyzed, there was a notable trend of decrease in the prevalence with the increase in sports activities intensity. We have found a significant drop in the trendline in subjects with more than 60 MET-h/w ($p = 0.032$) (Figure 1).

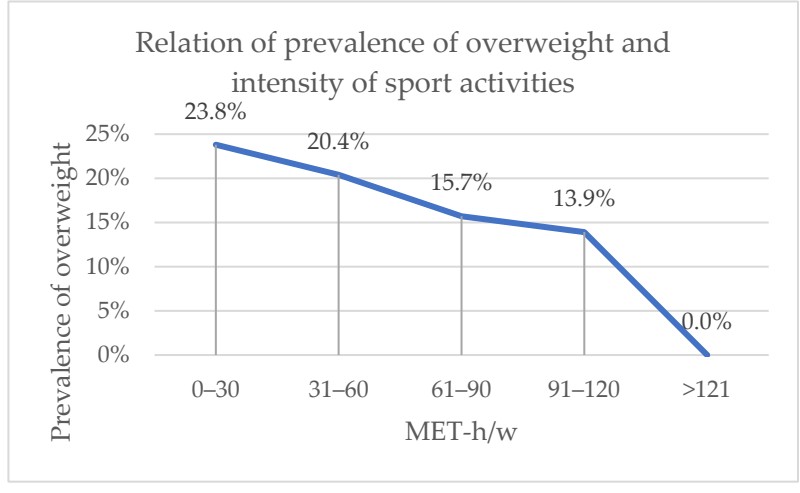

**Figure 1.** Prevalence of Overweight by MET-h/w Categories.

## 4. Discussion

In this study, we aimed to explore the body weight status of physically active and non-active children, as well as potential differences in the prevalence of overweight and obesity between different types and intensities of sports activities, in school children actively engaged in various sports activities.

Children in Serbian primary schools are having only two school classes of 45 min of physical education per week, a total of 4.8 h of physical activities per week on average [2], which is under the recommended level of 60 min of physical activity per day, as recommended by WHO [29]. To increase activity level and energy expenditure, there were numerous attempts, either in school-based programs or within extracurricular activities. Regarding that, as shown in a meta-analysis by Harris et al. [12], school-based physical activity interventions are not as effective in improving BMI. On the other hand, extracurricular physical activities, such as involvement in youth sports can contribute by increasing energy expenditure. Katzmarzyk and Malina determined that engagement in youth sports activities increases total daily energy expenditure by 20% [30], or by 30 min (11 moderate, 19 vigorous) in terms of additional time of MVPA (Moderate-to-Vigorous Physical Activity) per day, as revealed by Wickel and Eisenmann [31].

This research confirmed the hypothesis that a lower prevalence of overweight is found in children involved in organized sports activities compared to non-active children, similar to another study in Serbia [32], as well as national studies conducted in Spain [13], Denmark [19], the United States [14], and Greece [16]. A meta-analysis by Kim et al. [22] has shown that sport-based intervention has moderate but positive effects on the body weight status of children. However, a recent meta-analysis of Oliveira [23], as well as some other studies [15,17,20,21,24], did not find a clear association between sports participation and the body weight status of children. Cairney and Veldhuizen suggest that the relationship between sports and body weight is present but only becomes important at levels of activity higher than those reported in the abovementioned studies [33,34], which is in concordance with our results saying that only larger numbers of hours of training per week and/or years of training are effective in maintaining a normal body weight status.

We were somewhat surprised not to find a clear relationship between the type of sports activities and body weight status, as we were expecting that endurance sports, as well as mixed sports, should be more effective in maintaining a good body weight status, as reported in some papers [35,36].

Lastly, we have found a positive relationship between the high intensity of sports activities expressed in energy expenditure according to the Youth Compendium of Physical Activities (in MET-hours per week) and the prevalence of overweight in children: the more MET-hours of activity per week the less is the prevalence of overweight. We have identified the value of 60 MET-h/w to be the breaking point from which the prevalence started to decline significantly.

The present cross-sectional study lacks insight into how a multitude of years of sports activities affects the body weight status of a child. For that reason, we shall be continuing to study the relationship between different types and intensities of sports activities and body weight status, aiming to follow the same subjects through many years of sports activities in a longitudinal study.

We are fully aware that other major factors are influencing the body weight status of children involved in youth sports, such as genetics, nutrition, processes of maturation, sex and metabolic hormones, other extracurricular activities, etc., but these are subjects of future investigations in this field.

## 5. Conclusions

This study supplies valuable insight into the relationship between sports activities, their different types and intensities, and the body weight status of school children, which significance is beyond Serbian national scope.

Namely, we have found that type of sports does not play a significant role in the prevention of overweight in children.

Our finding that sports activities more intensive than 60 MET-h/w are associated with a lower prevalence of overweight in children is, de facto, one of the first quantified results in children's overweight studies.

In the end, this study adds to the body of evidence that regular involvement in sports activities presents one of the major contributing factors in the sustainable prevention of overweight or obesity.

**Author Contributions:** Conceptualization, S.M. (Sead Malićević) and S.M. (Sanja Mazić); methodology, S.M. (Sead Malićević) and G.G.; software, S.M. (Sead Malićević); validation, S.M. (Sead Malićević), S.M. (Sanja Mazić) and G.G.; formal analysis, S.C.; investigation, S.M. (Sead Malićević) and S.M. (Sanja Mazić); resources, G.G. and F.F.; data curation, S.M. (Sead Malićević) and G.G.; writing—original draft preparation, S.M. (Sead Malićević) and S.M. (Sanja Mazić); writing—review and editing, S.M. (Sead Malićević), G.G., F.F. and S.C.; visualization, F.F.; supervision, S.M. (Sanja Mazić); project administration, S.C. All authors have read and agreed to the published version of the manuscript.

**Funding:** This research received no external funding.

**Institutional Review Board Statement:** The study was conducted in accordance with the Declaration of Helsinki and approved by the Ethics Committee of the School of Medicine, the University of Belgrade, Serbia (protocol code 2650/IV-5, dated 10 April 2018).

**Informed Consent Statement:** Informed consent was obtained from all parents or legal guardians of the subjects involved in the study.

**Data Availability Statement:** The data presented in this study are available on request from the first author. The data are not publicly available due to privacy.

**Conflicts of Interest:** The authors declare no conflict of interest.

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
