# Peer review of "Relationship between Type and Intensity of Sports Activities and the Prevalence of Overweight in Serbian School Children"

_sustainability, doi:10.3390/su14137978_

Round 1

Reviewer 1 Report

There are still in title and abstract sentences that suggest "effects" or "influences" of intensities of activities on prevalence of overweight:

Title must be changed  as follows: Relationship between type.... and the prevalance of overweight....

In abstract:

line 12...to explore the relationship between..

line 2 a significant relationship..

Maybe it is good to include in the discussion a sentence about the restrictions of the crosssectional study to conclude about effects and influence of their independent and dependent variables.

Reviewer 2 Report

I read with attention the present study investigating the associations between the nature of sport and intensity of sport activity in obese/overweight children. Overall, the rationale for performing this study is well explained and the results are interesting. I believe that present article has merit but I have some major comments that should be addressed.

Introduction:

Lines 51-53: please consider to expand the relationship between body composition and sport performance (i.e. combact sports Vs. team sports Vs. individual sports).

Line 63: before describing the main purposes of the study, please consider to explore the importance of movement quality-training and motor competence as factors that could improve sport commitment, health and wellbeing in both adults and children with obesity.

Here some references to consider:

Trecroci A, Invernizzi PL, Monacis D, Colella D. Actual and Perceived Motor Competence in relation to Body Mass Index in Primary School-Aged Children: A Systematic Review. Sustainability 2021, 13, 9994.

Cavaggioni L, Gilardini L, Redaelli G, Croci M, Capodaglio P, Gobbi M, Bertoli S. Effects of a Randomized Home-Based Quality of Movement Protocol on Function, Posture and Strength in Outpatients with Obesity. Healthcare 2021, 9, 1451.

Material and Methods

Line 73: The section “participants and procedures” needs a major clarification.

Line 114: in addition to anthropometric parameters, did the Authors also measured the maturity status?

Discussion

Line 178: Please consider to elaborate if the results obtained by the Authors meet the international guidelines regarding the minimum quantity of physical activity (Chaput JP, Willumsen J, Bull F, Chou R, Ekelund U, Firth J, Jago R, Ortega FB, Katzmarzyk PT. 2020 WHO guidelines on physical activity and sedentary behaviour for children and adolescents aged 5-17 years: summary of the evidence. Int J Behav Nutr Phys Act. 2020 Nov 26;17(1):141).

Line 208: Line 208: Please consider to clarify the main limitations of the current study.

Round 2

Reviewer 2 Report

I don't have further comments or suggestions for the Authors

This manuscript is a resubmission of an earlier submission. The following is a list of the peer review reports and author responses from that submission.

Round 1

Reviewer 1 Report

- Introduction 

  1. This study aimed to identity the relation of nutritional status, and type and intensity of sports activities on the prevalence of overweight in Serbian children. However, as one of the readers, I don't see any a conceptual framework or model for the logical link among variables. To be specific, the authors should add evidences supporting association between nutritional status and overweight in children.
  2. Also, detailed reasons are required why the variables including sport type and intensity are involved in this study. In addition, the importance of this study should be investigated more based on the information or results that this study has indicated. 

- Materials and Methods

  1. Regarding sport classifications, the groupings are ambiguous and confused to proceed statistical analysis. I'd like to suggest that the classifications be simple, clear and general.
  2. This study presents basic statistics indicating only simple information. There needs to be undertaken with advanced statistics to identify improvements to previous studies. 

- Results

  1. This study focuses on the descriptions of some Tables presenting various classifications. It could be wondered if the logical connections are provided from research purpose to results.

Reviewer 2 Report

The aim of this manuscript is to prove that physical active children have lower overweight/obesity than physical inactive children and also which physical activity (intensity and duration) are better in this respect.

However in the paper (in titlle, abstract and throughout the full text)it is suggested that the differences are caused by physical activity. However in this crosssectional design any influence can not be proved.

Before making more detaied comments I wait for the resubmitted manuscript.

The conclusion is not based on their results and speculative.

Moreover most of the tables are not necessary (2,3,4 are published elsewhere and can be cited).

Also table 5 is making three age groups, but the important biological age is interfering these calendar age groups with respect to BMI status.

The term nutritional status is misleading because only BMI is measured.

Is grouping on the basis of weight and height  based on research?